# Publication outcomes and research trends in bachelor's and master's theses in health sciences in Croatia: retrospective cohort study and survey of mentors

Kristina Kraljic[1], Vesna Mijoc[1], Marin Cargo[1], Ognjen Barcot[2,3], Marta Civljak[1], Mario Marendic [4], Damir Sapunar[2], Andrea Vuksa[5], Tea Kabic[6], Darko Novak[7], Jelena Medakovic[1], Kata Ivanisevic [8], Zrinka Puharic[9,10], Natasa Skitarelic[11], Dijana Majstorovic [12], Marijana Neuberg[13], Snjezana Cukljek[8,14], Aleksandar Racz[8,14], Sanja Zoranic[15], Livia Puljak [1]*

1 Centre for Evidence-Based Medicine and Health Care, Catholic University of Croatia, Zagreb, Croatia, 2 University of Split School of Medicine, Split, Croatia, 3 Surgery Department, University Hospital of Split, Split, Croatia, 4 Faculty of Health Sciences, University of Split, Split, Croatia, 5 Department of Health Studies, University of Applied Sciences, Šibenik, Croatia, 6 Institute of Emergency Medicine of Split-Dalmatia County, Split, Croatia, 7 Health Center of the Ministry of the Interior of the Republic of Croatia, Zagreb, Croatia, 8 Faculty of Health Studies, University of Rijeka, Rijeka, Croatia, 9 Department of Nursing, Bjelovar University of Applied Sciences, Bjelovar, Croatia, 10 Faculty of Dental Medicine and Health, Josip Juraj Strossmayer University of Osijek, Osijek, Croatia, 11 Department of Health Studies, University of Zadar, Croatia, 12 Medical School, Juraj Dobrila University of Pula, Pula, Croatia, 13 Department of Nursing, University North, Varaždin, Croatia, 14 University of Applied Health Sciences, Zagreb, Croatia, 15 Department of Nursing, University of Dubrovnik, Dubrovnik, Croatia

* livia.puljak@gmail.com, livia.puljak@unicath.hr

## Abstract

### Objectives

Conducting original research within bachelor's and master's theses (dissertations) and publishing thesis results in scientific journals has multiple advantages. This study aimed to analyze types and publication outcomes of bachelor's and master's theses defended by students of health sciences studies in Croatia.

### Design

We conducted a retrospective cohort study of bachelor's and master's theses of students of health studies in Croatia, defended by May 14, 2022, and a cross-sectional study of mentors who were surveyed for publication output.

### Setting

We analyzed theses from repositories. Thesis mentors were surveyed via e-mail about the publication of the thesis content in scholarly journals. Additionally, three online sources (PubMed, Google Scholar, and Google) were used to find theses-based publications.

**Data availability statement:** The raw dataset collected within the study contains personal identifiers (students' names) and the full titles of the theses analyzed. Even with direct identifiers removed, thesis titles enable the identification of students. Complete anonymization is thus not feasible. The study protocol was approved by the Ethics Committee of the Catholic University of Croatia, which did not permit public data sharing because the individuals whose theses were analyzed did not provide consent for disclosure of their personal data. The dataset is also subject to the General Data Protection Regulation (EU GDPR). Accordingly, the raw data are not included in the manuscript. Data may be available from the corresponding author upon reasonable request, contingent on approval of the data access request by the Ethics Committee. Requests should be directed to the principal investigator, Prof. Livia Puljak (livia.puljak@unicath.hr), and to the Ethics Committee of the Catholic University of Croatia (eticko.povjerenstvo@unicath.hr).

**Funding:** The author(s) received no specific funding for this work.

**Competing interests:** The authors have declared that no competing interests exist.

**Abbreviations:** EBP, evidence-based practice; ICBAS, Catholic University of Lisbon and Institute of Biomedical Sciences Abel Salazar; JIF, journal impact factor; MOZVAG, the list of accredited study programs in Croatia; NSK, National and University Library of Croatia; UAB, Universitat Autònoma de Barcelona; VBA, visual basic for applications; ZIR, Croatian National Repository of Bachelor's and Master's theses.

## Participants

We surveyed mentors of bachelor's and master's theses of health sciences students in Croatia.

## Results

We analyzed 9861 theses, 7979 (81%) bachelor's and 1882 (19%) master's theses. Most (66%) of master's theses were based on original research, compared to 21% of bachelor's theses. Mentors of 3910 (40%) theses responded to the survey, indicating that 176/3910 (4.5%) of theses results were published in scientific journals. Through additional search of online sources, we found that from all analyzed theses, 276 (2.8%) articles were published in a scholarly journal. Among published articles, 229 (83%) were published in Croatian journals.

## Conclusions

Most analyzed theses were not based on original research. Results of a few theses were published in a scientific journal. Institutions should encourage mentors and students to conduct original research within theses and to publish thesis results in a journal, to further develop health sciences.

## Introduction

At the end of many bachelor's and master's study programs, the student obtains a degree after preparing a thesis (dissertation) that is reviewed and evaluated by experts in a specific field. The purpose of preparing a thesis in health sciences studies is to train healthcare workers to better understand the values of scientific research for their career development. It has been shown that writing a bachelor's thesis is associated with greater interest and a favourable attitude towards nursing research and development [1]. Furthermore, recent evidence shows that writing the bachelor's thesis measurably improves nursing students' evidence-based practice (EBP), information-retrieval, and application skills [2].

A study focusing on master's degree research among allied health professionals found that completing a master's dissertation had a positive impact on individual learning outcomes such as confidence, clinical practice capability, and research skills. However, formal research dissemination from these theses was low, but ongoing research engagement was high. The study concluded that master's degree training is an under-recognized source of research capacity building in allied health professions, even if broader clinical service impacts remain unclear. The research questions were mostly self-generated, and the study suggested that more coordinated research prioritization could improve broader impact [3].

A recent scoping review of the bachelor's thesis in nursing education further emphasizes that, despite its pedagogical value, there remain substantial gaps in how theses are structured, supervised, and evaluated, underscoring the need for clearer educational frameworks and guidance for both students and faculty [4].

Multiple studies in the literature have explored theses defended by health sciences students. A study conducted in Spain on 420 bachelor's theses of fourth-year nursing students carried out at the Universitat Autònoma de Barcelona (UAB) from 2013 to 2018 showed that all theses described a research proposal, similar to a grant proposal. Most theses described qualitative research (60%). A survey of students who prepared those theses indicated that most reported feeling satisfied with the knowledge and skills acquired during the thesis preparation [5].

Analysis of nursing master's theses (N = 273) defended from 2000-2010 at the *Catholic University of Lisbon* and *Institute of Biomedical Sciences Abel Salazar* (ICBAS) in Portugal showed that 59% used qualitative methods, 29% used quantitative methods and 5% used both. The authors identified deficiencies in the preparation and reporting of theses that could be improved [6].

More recent work highlights supervision as a key pedagogical mechanism during thesis courses and documents students' high expectations for group supervision and self-efficacy across the thesis process [7,8].

Higher education in Croatia is organized in line with the Bologna declaration [9]. Namely, the European Higher Education Area introduced a three-cycle higher education system consisting of bachelor's, master's and doctoral studies [9]. According to the Croatian Act on Higher Education and Scientific Activity, finalization of bachelor's and master's studies includes either the defense of a thesis or the passing of a final exam [10].

In Croatian health sciences studies, there are two distinct types of theses – one based on theoretical literature reviews (essay-type thesis) and the other based on original, empirical research. In master's programs, there is more emphasis on conducting a research-based thesis compared to bachelor's level. This differentiation reflects the educational objectives of fostering both theoretical knowledge and practical research skills among students [11]. The emphasis on original research in master's programs aligns with global trends in higher education that prioritize the development of research competencies as a means to enhance academic rigor and professional readiness [12]. A 2025 multi-stakeholder qualitative study further underscores how thesis work consolidates competencies for clinical practice and informs supervision models [13].

The publication outcomes of theses, i.e., whether they lead to publications, are critical indicators of academic success and impact. Research shows that students who engage deeply with their thesis work, particularly through original research, are more likely to produce publishable results [12]. This aspect is vital for advancing knowledge within the health sciences field and contributes to the overall quality of healthcare education in any given country.

There is a paucity of studies on theses defended in bachelor's and master's health sciences studies in Croatia. In 2020/2021, Marendic et al. surveyed 912 nursing students in Croatia to study attitudes and factors influencing the choice of thesis type, i.e., whether the students choose to conduct original research versus an essay-type literature review within their thesis. The study showed that modifiable factors, such as a mentor's encouragement, were associated with positive attitudes toward scientific research and preference for a research-based thesis among nursing students [11]. Those findings suggest that institutional support plays a crucial role in shaping students' academic trajectories. Furthermore, this underscores the need for effective mentorship programs within health sciences departments to enhance student engagement and success.

Among universities in Croatia, many undergraduate nursing programs require a bachelor's thesis, with literature reviews being a common format. However, there is a growing recognition of the value of original research as part of undergraduate education, reflecting an evolving landscape in health sciences education [11].

It is not known to what extent students of health sciences in Croatia conduct original studies as part of their bachelor's and master's theses. It is also not known to what extent results of bachelor's and master's theses of health sciences in Croatia are published in scholarly journals. The exploration of types and publication outcomes of bachelor's and master's theses in health sciences studies would be crucial for understanding educational effectiveness in Croatia. By examining these aspects, new research results can provide insights into how academic structures can better support students' research endeavors, ultimately contributing to improved health outcomes through enhanced educational practices.

Thus, this study aimed to analyze types and publication outcomes of theses defended by students of health sciences studies in Croatia.

## Methods

### Study design

This was a mixed-methods study, including a retrospective cohort study and a survey of mentors.

### Ethics

The study protocol was approved by the Ethics Committee of the Catholic University of Croatia (reference numbers: Classification number 641-03/22-03/024; Registration number 498-15-06-22-004). Written informed consent was obtained from all participants. Their confirmatory response to the invitation e-mail was considered as their informed consent to participate in the study, as indicated to the participants in the information sheet. All methods were carried out in accordance with relevant guidelines and regulations.

### Inclusion criteria

Bachelor's and master's thesis of health sciences students defended by May 14, 2022 available in the online National repository of bachelor's and master's theses ZIR (Croatian name of the repository: *Nacionalni repozitorij završnih i diplomskih radova ZIR*), were included in the analysis. Theses from all higher education institutions offering health sciences studies in Croatia were included in the analysis. We included all eligible theses from the ZIR, and those were published from 2011 to 2022.

Information about health sciences studies and institutions organizing them were retrieved from MOZVAG (https://www.azvo.hr/en/higher-education/mozvag-information-system), an information system developed by the Croatian Agency for Higher Education for accreditation purposes. We included theses from bachelor's and master's health sciences studies registered in the field of basic and clinical medical sciences.

The eligible institutions (N = 14) were: Catholic University of Croatia, Libertas International University, Josip Juraj Strossmayer University of Osijek, Juraj Dobrila University of Pula, University North, University of Dubrovnik, University of Rijeka, University of Split, University of Zadar, University of Zagreb, University of Applied Sciences "Lavoslav Ruzicka" in Vukovar, Bjelovar University of Applied Sciences, University of Applied Sciences Ivanic-Grad, and University of Applied Health Sciences. From those institutions, 42 study programs were eligible for inclusion.

### Information sources

We aimed to analyse theses available in the online National repository of bachelor's and master's theses ZIR, provided by the National and University Library of Croatia (NSK). The repository is available on this link: https://zir.nsk.hr/. Since each institution is responsible for uploading the theses and their meta-data into the repository, we contacted each eligible institution to verify the accuracy of information for their institution, i.e., whether all defended eligible theses from their institutions are available in the online repository. In case we get information from the institutions that some theses are missing in the repository, we asked the institutions to share them with us in electronic form. We recorded all communication and responses with the institutions in terms of the accuracy of the online repository and the outcomes of requests for thesis sharing.

We anticipated that all theses would be written in the Croatian language, but in case some theses were written in other languages, we included those theses as well.

### Extracting the online-available data about the theses

First, we used the advanced search method in the database from the web address https://zir.nsk.hr/ to select all eligible theses and export them as a file. Then, according to the web addresses of the theses, automated web-scraping was performed using Visual Basic for Applications (VBA) macro-commands so that the HTML content from each open web

address was copied to a separate Microsoft Excel worksheet. VBA macro-commands were then also used to extract the predicted data into an overview table of all theses. Finally, the person in charge of data processing firstly filtered and verified the eligibility of the theses, and then verified all data and manually updated it if there were any errors. The following data were parsed from the ZIR repository: name of the institution, name of the student, thesis title, name of the mentor, month and year of thesis defense, study program, research field and sub-field, type of the study [University study or professional/Polytechnic study], degree (bachelor's or master's), link to the full text of the thesis in the ZIR repository.

## Accessing the full text of the theses

We aimed to access the full text of the theses in electronic form from the ZIR repository and download each thesis for further analyses. We requested electronic versions of the theses from the institutions, if they were not available for free download from the ZIR.

## Data extraction from theses

The following data were extracted from each thesis: language of the thesis (Croatian or other – if other, which language), type of thesis: essay/review (original research not conducted; literature reviewed) or research-based thesis. For research-based theses, we extracted the following information: study design(s) reported in the thesis (e.g., cross-sectional, cohort, qualitative, etc.), our estimation of the thesis study design, unit of analysis (e.g., humans, animals, literature, etc.), and sample size for the unit of analysis.

## Publications stemming from the theses

Between September 2022 and May 2023, we contacted mentors of the theses included in the analysis to collect information about whether they published the content of the thesis as an article in a scholarly journal, submitted it, or intended to submit it to a scholarly journal. We also asked them about the reasons for not publishing the thesis content in a scholarly journal and whether students expressed any interest in publishing the thesis content. The text of the survey for mentors is in S1 File.

## Publication search

If we were unable to obtain information about the publication status of theses from mentors, we searched for potential articles online. We searched PubMed, Google Scholar, and Google to detect any articles published in scholarly journals stemming from the analyzed theses across disciplines and languages. We selected PubMed for its curated biomedical indexing and reliable author/title matching; Google Scholar to broaden coverage to regional and locally indexed journals and early online content not (yet) indexed in PubMed; and Google to capture journal websites, institutional pages, and bilingual variants (Croatian/English) and to mitigate issues with diacritics and transliteration in names and titles.

For the search, we used the Croatian and English titles of the thesis, thesis keywords, student's and/or mentor's name. Two authors conducted the search for each thesis, independently. In case of uncertainty whether an article stems from a thesis, the third author was involved in the discussion.

We excluded from consideration conference abstracts/presentations and cases when theses were published/indexed in secondary sources such as ResearchGate or Semantic Scholar. Such reports were not considered an article in a scholarly journal.

## Bibliographic characteristics of scholarly articles stemming from PhD theses

The following data were extracted from the scholarly articles published from the analysed theses: name of the first author, order of the authorship for a student (i.e., first author, second author, last author), order of the authorship for a mentor (i.e.,

first author, second author, last author), year of publication, journal name, Journal Impact Factor (JIF) for the latest year available at the time of data extraction.

## Statistics

Descriptive statistics, frequencies and percents; means (with standard deviations) or medians (with range) for JIF of the published articles.

## Results

### Analysis of theses

By May 14, 2022, 10668 theses from targeted institutions and targeted study programs were published in ZIR. Among them, 807 theses were not included in the analysis. Namely, the University of Dubrovnik (170 theses) could not provide the complete text of the theses. University of Applied Sciences "Lavoslav Ruzicka" in Vukovar (538 theses) and Faculty of Dental Medicine and Health Osijek (59 theses) did not accept the invitation to participate in the study. We also excluded duplicate entries, as some institutions had duplicate theses published in ZIR: University of Rijeka (2 theses), Faculty of Medicine Osijek (4 theses), University of Split (24 theses), University of Zadar (2 theses), Bjelovar University of Applied Sciences (1 thesis) and the University of Applied Health Sciences (7 theses).

We included in the analysis 9861 theses; 7979 of all theses (81%) were bachelor's theses, and 1882 (19%) were master's theses. The analyzed theses were defended at twelve institutions (Table 1).

The largest number of analyzed theses was defended at the University of Applied Health Sciences from Zagreb (47%). Most theses were defended in year 2021 (17%). The largest number of defended theses came from the nursing study program (57%) and in the field of clinical medical sciences (85%). More theses were defended at professional/polytechnic higher education institutions (71%) compared to University studies. Most of the analyzed theses were defended at the bachelor's study level (81%). Most theses (99.79%) were written in Croatian; the rest were written in English. For one thesis, it was reported in the ZIR that it was defended in the Chinese language, but upon checking, it was determined that this was a mistake and that it was written in the Croatian language (Table 1).

Based on our assessment, most of the analyzed theses were based on a literature review (70%), while the rest described original studies. In the majority of theses (70%), it was not stated what kind of study design was used in the thesis. When the study design was reported, the majority described it as a cross-sectional study (73%). According to our assessment of the study design, most of the analyzed theses were based on cross-sectional studies. Cohort studies and testing of an educational intervention were used the least. In most theses, the unit of analysis was humans (Table 2).

Among the analyzed theses, the majority (81%) were bachelor's theses, while the rest were master's theses (19%). Most of the bachelor's theses were based on a literature review (79%). On the contrary, most master's theses were based on original research (66%). The majority of both bachelor's and master's theses did not report the study design; when they did report it, the majority were described as cross-sectional. Based on our own assessment, among both bachelor's and master's theses, the majority used a cross-sectional design. A comparison of the characteristics of bachelor's and master's theses is shown in Table 3.

### Survey of mentors about the publication of theses in a journal

Mentors of the analyzed theses were contacted by e-mail to collect information about the publication/non-publication of thesis content in scientific journals. Surveys were sent to all mentors, and mentors of 3910 theses responded (40%). According to completed surveys, a journal article was published from 176/3910 (4.5%) of the theses for which mentors responded. Of the 3734 theses without a publication, the content of 34 (0.9%) theses was submitted to a

**Table 1. Characteristics of 9861 analyzed thesis of health sciences studies in Croatia.**

| Variables | N (%)* |
|---|---|
| **Institution** | |
| University of Applied Health Sciences | 4665 (47) |
| University North | 1331 (14) |
| University of Split | 992 (10) |
| University of Rijeka | 836 (8.5) |
| Bjelovar University of Applied Sciences | 603 (6.1) |
| Faculty of Medicine Osijek | 566 (5.7) |
| University of Zagreb | 328 (3.3) |
| University of Zadar | 257 (2.6) |
| Catholic University of Croatia | 191 (1.9) |
| Juraj Dobrila University of Pula | 50 (0.5) |
| University of Applied Sciences Ivanic-Grad | 21 (0.2) |
| Josip Juraj Strossmayer University of Osijek | 21 (0.2) |
| **Year of Thesis Defense** | |
| 2011 | 1 (0.01) |
| 2013 | 1 (0.01) |
| 2014 | 224 (2.3) |
| 2015 | 431 (4.4) |
| 2016 | 1193 (12) |
| 2017 | 1396 (14) |
| 2018 | 1610 (16) |
| 2019 | 1575 (16) |
| 2020 | 1627 (17) |
| 2021 | 1673 (17) |
| 2022 | 130 (1.3) |
| **Study Programme** | |
| Nursing | 5629 (57) |
| Physiotherapy | 2060 (21) |
| Medical – laboratory diagnostics | 708 (7.2) |
| Radiological technology | 554 (5.6) |
| Nursing – management in nursing | 389 (3.9) |
| Occupational therapy | 212 (2.2) |
| Midwifery | 196 (1.9) |
| Nursing – Promotion and protection of mental health | 56 (0.6) |
| Clinical nursing | 36 (0.4) |
| Psychiatric nursing | 21 (0.2) |
| **Field and subfield of research** | |
| Clinical medical sciences | 8403 (85) |
| Public health and health care | 918 (9.3) |
| Basic health sciences | 448 (4.5) |
| Pedagogy | 26 (0.3) |
| Educational and rehabilitation sciences | 20 (0.2) |
| Kinesiology | 19 (0.2) |
| Information and communication sciences | 11 (0.1) |
| Psychology | 4 (0.04) |
| Physics | 3 (0.03) |

*(Continued)*

**Table 1.** (Continued)

| Variables | N (%)* |
|---|---|
| Biotechnology | 2 (0.02) |
| Dental medicine | 2 (0.02) |
| Nutritionism | 2 (0.02) |
| Economy | 1 (0.01) |
| Pharmacy | 1 (0.01) |
| Veterinary medicine | 1 (0.01) |
| **Type of Study** | |
| Professional/Polytechnic | 7035 (71) |
| University | 2826 (29) |
| **Level of Study** | |
| Bachelor's | 7978 (81) |
| Master's | 1883 (19) |
| **Language** | |
| Croatian | 9840 (99) |
| English | 21 (0.21) |

*Percentages may not add up to 100 due to rounding.

journal. Most of them were submitted to the Croatian Nursing Journal (N = 8). Out of 3631 theses whose content was not submitted to any journal, 110 mentors indicated that they intended to prepare an article to be sent to a journal, mostly to a journal in the field of the scope of the thesis (N = 82). Also, 171 mentors stated that it does not matter to them to which journal they will send the thesis. The majority of mentors stated that lack of interest among students (N = 1234; 54%) and insufficient quality of theses (N = 437; 19%) were the main reasons for not publishing the thesis content in scientific journals. The mentors stated that 133 students expressed their interest in publishing their thesis. Some of the mentors stated that after the defense they wanted to publish the thesis, but the students no longer responded (N = 12) (Table 4).

### Literature search to find potentially published thesis papers

By combining the results of the mentor survey and the literature search, we found that based on 9861 theses, 276 (2.8%) articles were published that fully or partially correspond to the analyzed theses. In the majority of published journal articles (N = 182; 66%), a student was listed as the first author. The mentor was listed as the last author in 116 (42%) of the articles. Most articles were published in 2021 (N = 56; 20%). The largest number of journal articles stemming from the theses (N = 33; 12%) was published in the *Nursing Journal* (Croatian name: *Sestrinski glasnik*).

Journals in which five or more theses were published are listed in Table 5. The majority (229/276; 83%) of articles based on the theses were published in Croatian journals (Table 5).

The JIF for the year 2022 was available for 38 journals. The median of JIF of journals in which scientific articles based on theses were published was 2.3 (range: 0.133 to 8.1).

To facilitate quick overview of results, Table 6 consolidates the principal results reported above, including the study sample (N = 9,861 theses; institutions and study levels), overall and level-specific thesis types, our assessed distribution of study designs among original-research theses, units of analysis, mentor-survey response, and publication outcomes derived by integrating mentor reports with the literature search, including the proportion published in Croatian journals and the distribution of JIFs.

**Table 2. Type and study design of 9861 theses from health sciences studies in Croatia.**

| Variables | N (%)* |
|---|---|
| **Type of thesis** | |
| Literature review | 6949 (70) |
| Original research | 2912 (30) |
| **The structure of the research stated in the thesis (N=2912)** | |
| Not specified | 2029 (70) |
| Cross-sectional | 405 (14) |
| Case Report | 213 (7.3) |
| Retrospective | 197 (6.8) |
| Experimental study | 34 (1.2) |
| Matched-pairs study | 18 (0.6) |
| Prospective | 9 (0.3) |
| Cohort | 4 (0.1) |
| A randomized controlled trial | 3 (0.1) |
| **Our assessment of the research design (N=2912)** | |
| Cross-sectional | 2138 (73) |
| Retrospective | 440 (15) |
| Case Report | 215 (7.4) |
| Experimentally | 81 (2.8) |
| Matched-pairs study | 21 (0.7) |
| Prospective | 9 (0.3) |
| Cohort | 4 (0.1) |
| A randomized controlled trial | 3 (0.1) |
| Creating an educational intervention | 1 (0.03) |
| **Unit of analysis** | |
| Humans | 2373 (81) |
| Samples | 217 (7.4) |
| Documentation | 202 (6.9) |
| Animals | 61 (2.1) |
| Cells | 35 (1.2) |
| Literature | 10 (0.3) |
| Institutes | 4 (0.1) |
| Not specified | 3 (0.1) |
| Food | 3 (0.1) |
| Software | 1 (0.03) |
| Plasma | 1 (0.03) |
| Tissue | 1 (0.03) |

*Percentages may not add up to 100 due to rounding.

## Discussion

In a national cohort of 9861 health-sciences theses, most were literature reviews rather than original studies, especially at the bachelor's level, and only 2.8% of theses yielded a journal article. This quantifies a substantial evidence-to-publication gap across Croatian health sciences programs and provides a baseline for educational and policy reforms. Prior work links thesis-based original research with stronger EBP competencies; thus, curricula that increase original research

**Table 3. Comparison of the characteristics of bachelor's and master's theses.**

| Variables | Bachelor's theses, N (%)* | Master's theses N, (%)* |
|---|---|---|
| **Type of thesis** | | |
| Original research | 1672 (21) | 1241 (66) |
| Literature review | 6307 (79) | 641 (34) |
| **Study design reported in the thesis** | | |
| Not specified | 1187 (71) | 842 (68) |
| Case Report | 201 (12) | 12 (0.9) |
| Cross-sectional | 158 (9.5) | 248 (20) |
| Retrospective | 86 (5.2) | 111 (8.9) |
| Experimentally | 15 (0.9) | 19 (1.5) |
| Matched-pairs study | 11 (0.7) | 12 (0.9) |
| Prospective | 8 (0.5) | 7 (0.6) |
| A randomized controlled trial | 3 (0.2) | 1 (0.08) |
| Cohort | 3 (0.2) | 1 (0.08) |
| **Our assessment of study design** | | |
| Cross-sectional | 1155 (69) | 984 (79) |
| Retrospective | 245 (15) | 195 (16) |
| Case Report | 201 (12) | 14 (1.1) |
| Experimentally | 42 (2.5) | 39 (3.1) |
| Matched-pairs study | 14 (0.8) | 7 (0.6) |
| Prospective | 9 (0.5) | 0 (0) |
| A randomized controlled trial | 3 (0.2) | 0 (0) |
| Cohort | 3 (0.2) | 1 (0.08) |
| Creating an educational intervention | 0 (0) | 1 (0.08) |
| **Unit of analysis** | | |
| Humans | 1364 (82) | 1010 (81) |
| Samples | 126 (7.5) | 91 (7.3) |
| Documentation | 117 (7.0) | 86 (6.9) |
| Animals | 38 (2.3) | 23 (1.9) |
| Cells | 20 (1.2) | 15 (1.2) |
| Not specified | 3 (0.2) | 0 (0) |
| Literature | 2 (0.1) | 8 (0.6) |
| Food | 1 (0.06) | 2 (0.2) |
| Plasma | 1 (0.06) | 0 (0) |
| Institutes | 0 (0) | 4 (0.3) |
| Software | 0 (0) | 1 (0.08) |
| Tissue | 0 (0) | 1 (0.08) |

*Percentages may not add up to 100 due to rounding.

theses, strengthen supervision and research methods training, and incorporate explicit dissemination plans may translate to better bedside decision-making. Future studies should investigate whether changes, such as requiring original research for all types of theses and structured supervision, affect the dissemination output of health sciences theses.

These results point to specific actions for programs, requiring original research conducted for bachelor's and master's theses, strengthening structured supervision and writing support, and planning dissemination in a scholarly journal from

**Table 4. The results of the survey of mentors included in the analysis on the publication/ non-publishing of theses in scientific journals.**

| Theses publication | N (%)* |
|---|---|
| **The content of the bachelor's/master's theses was published in a scientific journal** | |
| No | 3734 (96) |
| Yes | 176 (4.5) |
| **The article based on the bachelor's/master's theses has not been published, but has been submitted to a journal** | |
| No | 3631 (99) |
| Yes | 34 (0.9) |
| **The journal to which the article was submitted** | |
| Croatian Nursing Journal | 8 (14) |
| Global Nursing and Healthcare | 6 (11) |
| Nursing Journal | 6 (11) |
| Croatian Journal of Health Sciences | 4 (7.0) |
| Journal of Health Sciences | 4 (7.0) |
| PHYSIOinfo | 2 (3.5) |
| Croatian Journal of Public Health | 2 (3.5) |
| Pediatria Croatica | 2 (3.5) |
| Medical Archives | 1 (1.8) |
| Works at HAZ | 1 (1.8) |
| Nurse Education Today | 1 (1.8) |
| Acta medica Croatica | 1 (1.8) |
| Physical and rehabilitation medicine | 1 (1.8) |
| Croatian Journal of Infection | 1 (1.8) |
| Medicus | 1 (1.8) |
| Education and research for quality healthcare practice | 1 (1.8) |
| Croatian Review of Rehabilitation Research | 1 (1.8) |
| Blue focus (Croatian: Plavi fokus) | 1 (1.8) |
| ST-Open | 1 (1.8) |
| Police and security | 1 (1.8) |
| World of Health | 1 (1.8) |
| European Journal of Orthodontics | 1 (1.8) |
| Journal of Hygienic Engineering and Design | 1 (1.8) |
| Documentation and measurement in physiotherapy | 1 (1.8) |
| Proceedings of the international congress | 1 (1.8) |
| Journal of the American Psychiatric Nurses Association | 1 (1.8) |
| Book of Abstracts | 1 (1.8) |
| The Journal of The Croatian Medical Association | 1 (1.8) |
| Annual of Social Work | 1 (1.8) |
| Medical Journal | 1 (1.8) |
| Journal of Community Health Nursing | 1 (1.8) |
| **The article has not been submitted, but it is intended to be prepared for a journal submission** | |
| No | 3330 (94) |
| Yes | 110 (3.1) |
| Maybe | 86 (2.4) |

*(Continued)*

**Table 4.** (Continued)

| Theses publication | N (%)* |
|---|---|
| **Target journals for article submission** | |
| Doesn't matter | 171 (48) |
| Journal in the subject area/certainly indexed | 82 (23) |
| We haven't decided | 35 (9.9) |
| PubMed, Scopus | 8 (2.3) |
| Journal of Applied Health Sciences | 6 (1.7) |
| Croatian Journal of Health Sciences | 6 (1.7) |
| Nursing Journal | 5 (1.4) |
| The Journal of The Croatian Medical Association | 5 (1.4) |
| Acta Clinica Croatica | 3 (0.9) |
| Scopus | 3 (0.9) |
| Collegium Anthropologicum | 3 (0.9) |
| Medica Jadertina | 3 (0.9) |
| Shock | 2 (0.6) |
| Nursing journal in the field of education | 2 (0.6) |
| National Nursing Journal | 2 (0.6) |
| BMC Medical Education | 1 (0.3) |
| International Journal of Clinical Pharmacy | 1 (0.3) |
| Midwife journal in the Republic of Croatia or one of the midwife journals | 1 (0.3) |
| Developmental Medicine & Child Neurology | 1 (0.3) |
| Journal in the field of physiotherapy/rehabilitation | 1 (0.3) |
| World of Health | 1 (0.3) |
| Croatian Nursing Journal | 1 (0.3) |
| Psychiatria Danubina | 1 (0.3) |
| Caediologia Croatica | 1 (0.3) |
| Research Notes | 1 (0.3) |
| Acta Clinica Pediatrica | 1 (0.3) |
| FEMS Microbes | 1 (0.3) |
| American Journal of Dance Therapy | 1 (0.3) |
| Journal of Physical Medicine and Rehabilitation | 1 (0.3) |
| Nurse Education in Practice | 1 (0.3) |
| Physiotherapy Croatica | 1 (0.3) |
| Lower ranked journals | 1 (0.3) |
| **The reason for not publishing the article** | |
| Lack of interest of the student | 1234 (54) |
| Insufficient quality | 437 (19) |
| Review work, little scientific value | 297 (138) |
| Lack of time | 199 (8.7) |
| Literature saturated with the topic of the title | 38 (1.7) |
| I don't know, maybe the student published the work without me | 28 (1.2) |
| Presented at the symposium | 26 (1.1) |
| The student refused | 16 (0.7) |
| COVID | 6 (0.3) |
| We have already published something similar earlier | 5 (0.2) |
| No interested magazine was found in the Republic of Croatia | 3 (0.1) |

*(Continued)*

**Table 4.** (Continued)

| Theses publication | N (%)* |
|---|---|
| **Did the student who defended the thesis express an interest in publishing the content of the thesis in a scientific journal?** | |
| No | 3178 (96) |
| Yes | 133 (4.0) |

*Percentages may not add up to 100 due to rounding.

the outset. On the practice side, more original, patient- and service-focused student projects should translate into stronger EBP skills and locally relevant evidence that can be incorporated into protocols and quality-improvement efforts.

In our sample of 9861 theses defended in health sciences studies in Croatia, only one-third were based on original research. An important interpretive point is the alignment between the level of education and the thesis type. At the master's level, theses were mostly based on original research (66%), whereas bachelor's theses were mostly literature reviews (79%). Yet even within original-research theses, cross-sectional designs predominated for both levels, indicating an opportunity to foster longitudinal, interventional, or mixed-methods projects, particularly at the master's level, where student capacity is higher.

Of note, our sample of theses was dominated by professional/polytechnic programs (71%) and bachelor-level theses (81%), with the largest single institutional contributor accounting for nearly half of all theses; programs in nursing (57%) and physiotherapy (21%) predominate. This should be considered when generalizing beyond Croatian health sciences programs.

Research should be the backbone of health sciences and should be advanced during higher education. By writing a literature review or an essay as part of a thesis, students of health sciences studies do not realize their full potential, and it is questionable how much such theses contribute to the development of the health profession.

Across all analyzed theses in our sample, literature reviews far outnumbered original studies (70% vs 30%). Where designs were stated, reporting was often incomplete ("not specified" 70%); where we could classify, cross-sectional designs dominated (73%) and truly experimental work was rare. This pattern signals both a capacity gap for primary research and a reporting gap that programs could address in future interventions.

Developing and sustaining research initiatives among undergraduate and graduate students can benefit academic institutions, faculty mentors, and students. More research is needed to advance knowledge and innovation in all fields of medicine. This implies that students must be prepared for today's knowledge-driven world. The main goals of research methodology education at the undergraduate and graduate level are to teach students how to plan, conduct and report a study. The key to successful participation in research is for students to see and understand the importance of rigor, academic integrity and responsible research conduct. This means that academic institutions should carefully plan research programs, activities and education in research methodology for students. Building capacity for research has a long-term impact on valuable learning outcomes as undergraduate and graduate students prepare for professional service. Therefore, it is extremely important to strengthen the capacities for the involvement of students in original research [14].

We were not able to find any prior studies in the literature about the publication of results of bachelor's and master's theses from health sciences studies in Croatia, so we can not compare our results with similar studies in the same setting. Comparison can be drawn with similar studies in other countries. A study conducted by Roca et al. in Spain, which analyzed all defended bachelor's theses in nursing studies in the period from 2015 to 2016, showed that 68% of the bachelor's theses were based on original research [15]. According to Baggio et al., who studied nursing studies in Portugal from 2000 to 2010, out of 273 master's theses, 210 (77%) were based on original research [6].

**Table 5. Analysis of 276 journal articles stemming from theses of health sciences studies in Croatia.**

| Variables | N (%)* |
|---|---|
| **Order of authorship for the student** | |
| 1. | 181 (66) |
| 2. | 52 (19) |
| Student not an author | 11 (3.9) |
| last | 11 (3.9) |
| 3. | 10 (3.6) |
| 4. | 3 (1.1) |
| 5. | 3 (1.1) |
| 6. | 2 (0.7) |
| 7. | 1 (0.3) |
| 8. | 1 (0.3) |
| **Order of authorship for the mentor** | |
| last | 116 (42) |
| 1. | 66 (24) |
| Mentor not an author | 43 (16) |
| 2. | 41 (15) |
| 3. | 6 (2.1) |
| 5. | 2 (0.7) |
| 7. | 2 (0.7) |
| **Year of publication** | |
| 2021. | 56 (20) |
| 2022. | 54 (20) |
| 2020. | 46 (17) |
| 2019. | 41 (15) |
| 2018. | 31 (11) |
| 2017. | 19 (6.9) |
| 2023. | 14 (5.1) |
| 2016. | 11 (5.1) |
| 2015. | 5 (1.8) |
| **Journal where the article was published**** | |
| Nursing Journal (Sestrinski glasnik) | 33 (12) |
| Croatian Nursing Journal | 20 (7.2) |
| Journal of applied health sciences | 15 (5.4) |
| Midwifery Journal (Primaljski vjesnik) | 14 (5.1) |
| Acta medica Croatica | 13 (4.7) |
| Croatian Journal of Health Sciences (Hrvatski časopis zdravstvenih znanosti) | 12 (4.3) |
| Southeastern European Medical Journal | 10 (3.6) |
| Medicina Fluminensis | 9 (3.2) |
| Medica Jadertina | 9 (3.2) |
| Physiotherapia Croatica | 6 (2.1) |
| Food in health and disease (Hrana u zdravlju i bolesti) | 6 (2.1) |
| Journal of Hygienic Engineering and Design | 6 (2.1) |
| ERS – Edukacija, Rekreacija, Sport | 5 (1.8) |
| Teacher review (Nastavnička revija) | 5 (1.8) |
| Safety (Sigurnost) | 5 (1.8) |

*(Continued)*

**Table 5.** (Continued)

| Variables | N (%)* |
|---|---|
| World of health | 5 (1.8) |
| **Published in a Croatian journal** | |
| Yes | 229 (83) |
| No | 47 (17) |

*Percentages may not add up to 100 due to rounding.

**Journal name showed if it published≥5 articles from the sample.

**Table 6. Key results at a glance – summary of sample, thesis characteristics, and publication outcomes.**

| Domain | Metric | Value |
|---|---|---|
| **Corpus** | Theses analyzed; institutions | **9861** theses; **12** institutions. |
| | Degree level | Bachelor's **7979 (81%)**; |
| | | Master's **1882 (19%)**. |
| **Thesis type** | Overall | Literature review **6949 (70%)**; Original research **2912 (30%)**. |
| | By level | **Bachelor's** original research **1672 (21%)** vs literature review **6307 (79%)**; |
| | | **Master's** original research **1241 (66%)** vs literature review **641 (34%)**. |
| **Study design (our assessment; among original-research theses, n=2,912)** | Cross-sectional | **2138 (73%)**. |
| | Retrospective | **440 (15%)**. |
| | Case report | **215 (7.4%)**. |
| | Other designs (each ≤3%) | Experimental **81 (2.8%)**; |
| | | Matched-pairs study **21 (0.7%)**; Prospective **9 (0.3%)**; |
| | | Cohort **4 (0.1%)**; |
| | | Randomized controlled trial **3 (0.1%)**; |
| | | Educational intervention **1 (0.03%)**; |
| | | |
| **Unit of analysis (original-research theses)** | Humans | **2,373 (81%)** (others each ≤8%). |
| **Language** | Thesis language | **99.79% Croatian**, remainder English. |
| **Mentor survey** | Responses | **3910 (40%)** of theses had mentor responses. |
| | Articles reported published (within respondents) | **176/3,910 (4.5%)**. |
| **Overall publications (survey+search)** | Articles published tied to theses | **276/9861 (2.8%)**; |
| | | student first author **181 (66%)**; |
| | | mentor last author **116 (42%)**. |
| | Journals and indexing context | **83% (229/276)** in Croatian journals; median JIF **2.3** (range **0.133–8.1**). |

Gros-Naves et al. reported that out of 204 surveyed graduate nurses in Spain, 12% reported that they had published the results of their master's theses in a scientific journal. Although nurses showed a positive attitude towards research, they also highlighted a lack of support and opportunities, resulting in a low number of published thesis articles. Gros-Naves et al. showed that students who wrote a bachelor's thesis expressed a greater willingness to engage in research when writing a thesis and the development of nursing compared to those who did not write a bachelor's thesis [1].

Beyond nursing, analyses in the field of physiotherapy showed that moving bachelor's/master's theses into journal articles remains challenging; in Brazilian physiotherapy programs 43% of undergraduate theses were disseminated, mostly in national journals (18%), and rarely in international journals (9%) [16].

Writing a thesis is challenging for most students; sometimes, they encounter research work for the first time, and at other times, they are insufficiently informed and educated about it. In Sweden, since 2007, writing a thesis has been a mandatory part of the classes that students enrolled in the nursing program attend during their studies at the university. Three categories that students must master are assessed: knowledge and understanding, competencies and skills, and judgment and approach [17].

Obuku et al. analyzed 1172 master's theses at the Makerere University College of Health Sciences in Uganda published over a period of 20 years. A manual search of Google Scholar and PubMed revealed that journal articles were published based on 209 (18%) analyzed master's theses [18].

Compared to these results, the proportion of articles based on health sciences studies theses in our study is smaller. The survey of mentors indicated that an article was published in a journal based on 4.5% of theses. The analysis of the literature for all theses in our sample, combined with the survey of mentors, showed that 276 (2.8%) theses resulted in a published journal article. Health sciences studies in Croatia should engage more in encouraging mentors and students to publish articles in journals based on defended theses. Otherwise, the results of research conducted within the framework of theses remain so-called grey literature that has not been published in a journal and is hardly accessible to international readers. Even if the thesis is publicly available on the Internet, most of the analyzed theses were written in Croatian, with only a summary written in English. Scientists from the rest of the world are unlikely to search Croatian thesis repositories to find relevant scientific literature.

Among mentors responding (40%), only 4.5% reported a published article; submissions in progress were rare (0.9%). The leading barriers were student lack of interest (54%) and insufficient thesis quality (19%), pointing to modifiable factors (motivation, writing/statistical support, authorship planning) rather than non-modifiable constraints.

Overall, among 2.8% theses that led to a journal article, students were first authors in 66% and mentors last authors in 42%. Also, it should be emphasized that 83% of outputs in articles appeared in Croatian journals, with potentially limited international reach. Strengthening English-language editing, journal targeting, and indexing coverage could raise discoverability and impact.

Our findings are consistent with a recent scoping review of bachelor's theses in nursing education, which identified widespread gaps in supervision models, evaluation frameworks, and institutional guidance. Addressing these issues through more structured support for both students and mentors could improve the quality and dissemination of thesis-based research [4].

Health workers are expected to be familiar with EBP. Asking clinical questions, conducting a systematic literature search, and conducting a critical appraisal of research results were some of the obstacles cited towards EBP. Nordsteien et al. evaluated how a collaborative intervention between the library and faculty teaching influenced the research skills of nursing students when writing bachelor's theses in Norway [19]. Quantitative and qualitative data were collected from the final year of a nursing degree program. Students were exposed to the intervention for one, two and three years in that period. Research skills related to EBP clearly improved over three years. There was an increase in the use of most EBP tools. Grades in the upper half of the grading scale increased from 67% to 82% from 2013. to 2015, and a positive correlation was found between grades and critical evaluation skills. The library-faculty collaborative teaching intervention used was successful in promoting nursing students' research skills regarding EBP principles [19].

In addition to interventions that will foster students' interest in research, institutional interventions can be implemented to ensure that more students conduct original research within their theses. Currently, multiple institutions where the analysed theses were defended allow students to conduct literature reviews instead of original research within their master's theses. The institutions could change their bylaws and require all students to conduct original research within master's theses.

Additionally, one could question the utility of writing theses based on literature reviews. The majority of theses we analyzed were based on a literature review/essay. Such theses may burden institutions and do not allow available mentors to focus on research work with students. Perhaps students and mentors take the line of least resistance, and they decide it is easier to write an essay/review thesis. One of the institutions that took part in this study, namely the Catholic University of Croatia, introduced the final exam instead of a bachelor's thesis in the academic year 2019/2020. However, considering the value of research-based theses for the development of future health sciences professionals, this decision may not be an ideal solution because research training is also an essential component of university education. Nevertheless, master's theses at this institution must be based on original research and follow strict methodological standards.

This study has several notable strengths. First, it included a very large number of theses (N = 9861) from multiple higher education institutions across Croatia, providing a comprehensive overview of thesis characteristics and publication outcomes in health sciences. Second, we applied a dual approach to assessing publication output by surveying mentors and performing searches across three information sources (PubMed, Google Scholar, and Google). This triangulation reduced the likelihood of missing published articles. Third, by combining institutional verification with repository screening, we ensured that data collection was robust and that our dataset captured a wide representation of defended theses in the country. Finally, our study provides the first systematic evidence on thesis outcomes in Croatian health sciences education, filling an important gap in the literature and offering a foundation for future educational and policy improvements.

This study also had several limitations. Although we included nearly ten thousand theses, some defended theses were unavailable in electronic format and therefore could not be analyzed. In addition, not all institutions agreed to participate, and not all mentors could be reached, which resulted in missing data. The mentor survey response rate was 40%, meaning that publication rates based solely on mentor reporting may be underestimated or incomplete. Furthermore, our publication search strategy was limited to three information sources (PubMed, Google Scholar, and Google). We did not search additional specialized databases, such as CINAHL, and the search was not conducted in a fully systematic manner, which may have led to missed publications. Finally, most theses were written in Croatian, which may have restricted their discoverability and visibility in international contexts. These limitations should be considered when interpreting the findings.

### Suggestions for future research and practice

It would be useful to repeat this study after several years to determine whether the number of theses on health sciences studies that are based on original research and published in journals as articles will change in future. It would also be useful to examine interventions that can help encourage students and mentors to publish thesis content in scientific journals. Furthermore, for part of the theses that were included in ZIR, the full texts of the theses are not publicly available, so we obtained such theses directly from the institutions that were involved in the study. The recommendation for the future would be that all theses defended at universities in Croatia should be published in ZIR in full text so that they are available to the public for reading and evaluation.

### Conclusion

In a national cohort of 9,861 health sciences theses, 70% were based on a literature review (70%), while the rest described original studies, and results of only 2.8% theses were disseminated as a journal article. Our estimates should be interpreted in light of several limitations, as the mentor survey response was 40%, some defended theses were not available in electronic form, and our publication search was limited to three information sources. These results quantify a substantial thesis-to-publication gap. Sustained improvement will require a deliberate shift from essay-based theses to empirical studies, coupled with systematic dissemination in scholarly journals, so that student research can credibly inform clinical practice and health policy.

## Strengths and limitations of this study

- A large number of theses were included in the analysis (N = 9861), and analysis of published articles was done by surveying mentors and searching three electronic information sources.

- Some theses defended at eligible institutions were not available for analysis as they were not converted to electronic formats, and were not available in the national or institutional repository.

- Only theses from institutions currently offering health sciences studies were included; theses from institutions that offered such programs in the past were excluded.

- Not all mentors responded to the survey; some could not be reached as their email addresses were no longer functional, they were not in the active workforce anymore, and some had deceased.

- Not all invited institutions accepted to participate, reducing the amount of data collected.

## Supporting Information

**S1 File.  Text of the survey for mentors.**
(DOCX)

## Author contributions

**Conceptualization:** Marta Civljak, Damir Sapunar, Livia Puljak.

**Data curation:** Kristina Kraljic, Ognjen Barcot, Andrea Vuksa, Tea Kabic, Darko Novak, Jelena Medakovic, Kata Ivanisevic, Natasa Skitarelic, Dijana Majstorovic, Marijana Neuberg, Snjezana Cukljek, Aleksandar Racz, Sanja Zoranic, Livia Puljak.

**Formal analysis:** Kristina Kraljic, Vesna Mijoc, Marin Cargo, Darko Novak, Zrinka Puharic, Livia Puljak.

**Investigation:** Kristina Kraljic, Vesna Mijoc, Marin Cargo, Ognjen Barcot, Marta Civljak, Mario Marendic, Damir Sapunar, Andrea Vuksa, Tea Kabic, Darko Novak, Jelena Medakovic, Kata Ivanisevic, Zrinka Puharic, Natasa Skitarelic, Dijana Majstorovic, Marijana Neuberg, Snjezana Cukljek, Aleksandar Racz, Sanja Zoranic, Livia Puljak.

**Methodology:** Ognjen Barcot, Marta Civljak, Mario Marendic, Damir Sapunar, Andrea Vuksa, Kata Ivanisevic, Livia Puljak.

**Project administration:** Livia Puljak.

**Supervision:** Livia Puljak.

**Writing – original draft:** Kristina Kraljic, Vesna Mijoc, Marin Cargo, Ognjen Barcot, Marta Civljak, Mario Marendic, Damir Sapunar, Andrea Vuksa, Tea Kabic, Darko Novak, Jelena Medakovic, Kata Ivanisevic, Zrinka Puharic, Natasa Skitarelic, Dijana Majstorovic, Marijana Neuberg, Snjezana Cukljek, Aleksandar Racz, Sanja Zoranic, Livia Puljak.

**Writing – review & editing:** Kristina Kraljic, Vesna Mijoc, Marin Cargo, Ognjen Barcot, Marta Civljak, Mario Marendic, Damir Sapunar, Andrea Vuksa, Tea Kabic, Darko Novak, Jelena Medakovic, Kata Ivanisevic, Zrinka Puharic, Natasa Skitarelic, Dijana Majstorovic, Marijana Neuberg, Snjezana Cukljek, Aleksandar Racz, Sanja Zoranic, Livia Puljak.

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
