## [Decision Letter · Decision Letter 0]

19 Aug 2025

Dear Dr. Puljak,

Thank you for submitting your manuscript to PLOS ONE. After careful consideration, we feel that it has merit but does not fully meet PLOS ONE’s publication criteria as it currently stands. Therefore, we invite you to submit a revised version of the manuscript that addresses the points raised during the review process.

We look forward to receiving your revised manuscript.

Kind regards,

Iskra Alexandra Nola

Academic Editor

PLOS ONE

Journal Requirements: 

3. Please remove all personal information, ensure that the data shared are in accordance with participant consent, and re-upload a fully anonymized data set.

Reviewers' comments:

Reviewer's Responses to Questions

**Comments to the Author**

1. Is the manuscript technically sound, and do the data support the conclusions?

Reviewer #1: No

Reviewer #2: Partly

2. Has the statistical analysis been performed appropriately and rigorously?

Reviewer #1: I Don't Know

Reviewer #2: Yes

3. Have the authors made all data underlying the findings in their manuscript fully available?

Reviewer #1: No

Reviewer #2: Yes

4. Is the manuscript presented in an intelligible fashion and written in standard English?

Reviewer #1: Yes

Reviewer #2: No

Reviewer #1: Thank you for the opportunity to review this interesting manuscript.

Some suggestions:

1. Overall, this is an interesting and necessary study.

2. The literature review requires updating with more recent studies.

3. The study’s aim is well-articulated, targeting a significant gap - Publication outcomes and research trends in Bachelor and Master theses in health sciences in Croatia.

4. Better to separate the Strengths and limitations of this study. For instance, the following is not clear if it is strength or limitation since the authors searched only 3 databases including Google and Google Scholar and did not include CINAHL and the search strategy was not clear and systematic.

49 - A large number of theses were included in the analysis (N=9861) and analysis of published

50 articles was done by surveying mentors and searching three electronic information sources.

5. In the introduction pages 68 and 69: you are limiting the interest of bachelor students to nursing research although in the title you mentioned “health sciences”

68 writing a bachelor thesis is associated with greater interest and a favorable attitude towards nursing

69 research and development.1

6. Create a concise results table:

Tighten your results with one clear table right after the narrative results so readers can scan the data fast.

7. Open the discussion with the main finding, then spell out its impact

One sentence: restate the top result.

Two to three sentences: explain how it shifts current evidence, guides future studies, EBP or changes bedside care.

Link results to implications for research and practice.

8. Rewrite the conclusion so it mirrors the results and discussion

Summarize the principal outcome and its practical meaning.

Echo any limits you noted earlier and close with a clear take-home message. Conclusion must reflect the evidence presented.

This paper has potential but needs a stronger analytical voice and updated evidence base to make a meaningful contribution.

Reviewer #2: 1. There are several grammatical errors in the manuscript. The authors should use a tool such as Grammarly to help them make corrections, such as Bachelor's and Master's theses in the manuscript.

2. The Discussion was particularly shallow for the huge data they gathered in 5 tables. There should have been at least one major takeaway from each of the tables. This is my major criticism of the manuscript.

3. The suggestion by the authors that the theses can be replaced by an examination is counterproductive. It is bad enough that many of their students are merely carrying out literature reviews instead of doing laboratory or field work; the authors should realize that research training is also an important component of university education.

**Do you want your identity to be public for this peer review?** For information about this choice, including consent withdrawal, please see our Privacy Policy

Reviewer #1: No

Reviewer #2: No

---

## [Author Response · Author response to Decision Letter 1]

18 Sep 2025

[Please note that we have also uploaded the Response as a separate file to preserve formatting.]

September 4, 2025

Re: Revision of the manuscript PONE-D-25-36339

Dear Editor,

We are very grateful for the editor's and reviewers' feedback regarding our manuscript titled „Publication outcomes and research trends in Bachelor and Master theses in health sciences in Croatia: retrospective cohort study and survey of mentors“ (PONE-D-25-36339).

Hereby, we are submitting a point-by-point response. In addition to this response, we are submitting a marked-up manuscript with revisions highlighted using track changes, and unmarked version of the revised manuscript.

Editor's comments:

1. I would like you to read the reviewers' comments in detail and to follow their suggestions thoroughly. It is important to note that in relation to the revision of the title, it is suggested to omit "health sciences" and add nursing, which I agree is more accurate.

Author response: We do understand that someone could have understood the results to be focused on nursing, because the majority of theses came from nursing students. However, 43% of analyzed theses were defended by students from other health sciences studies (other than nursing). Theses from all higher education institutions offering health sciences studies in Croatia were included in the analysis. We did not restrict our sample on theses from nursing study programs. This was indicated in the results: “The largest number of defended theses came from the nursing study program (57%).” Thus, since the sample of analyzed theses was not focused exclusively on theses defended by nursing students, we would kindly suggest keeping the title as it is.

2. Furthermore, I would like you to explain in more detail the selection of databases searched, to explain it following the remark of one of the reviewers.

Author response: We have now explained in more detail the selection of databases searched. We have expanded the Methods (Publication search subsection) to explain the rationale for the information sources chosen (PubMed, Google Scholar, Google), and how these sources complement each other for Croatian and English content. The following new text was inserted:

“We searched PubMed, Google Scholar and Google to detect any articles published in scholarly journals stemming from the analyzed theses across disciplines and languages. We selected PubMed for its curated biomedical indexing and reliable author/title matching; Google Scholar to broaden coverage to regional and locally indexed journals and early online content not (yet) indexed in PubMed; and Google to capture journal websites, institutional pages, and bilingual variants (Croatian/English) and to mitigate issues with diacritics and transliteration in names and titles.”

3. Please revise your conclusion following the suggestions given, especially since one of the reviewers highlighted the inconsistency between the topic and your proposal to replace the final paper with an exam.

Author response: We have now rewritten the conclusion. Also, we changed/updated the suggestion regarding the replacing the final paper with an exam. We have now simply mentioned that one of the institutions in our study replaced the bachelor thesis with an exam, but that this may not be the ideal solution because research training is an essential component of university education.

4. Overall, I find your paper very interesting, but I think it needs to be rewritten to give more context to your research and emphasize the difference in the type of thesis in relation to the level of education achieved, which could be considered somewhat inappropriate. You can also relate these findings to the education system in Croatia, if you feel that this is appropriate for an overall explanation or comment.

Author response: We appreciate the kind words of the editor. We have now:

-provided more context to our research,

-emphasized the difference in the type of thesis in relation to the level of education achieved, and

-provided context of our findings in the education system in Croatia.

Reviewers' comments:

Reviewer #1:

1. Overall, this is an interesting and necessary study.

Author response: We appreciate the kind words of the reviewer.

2. The literature review requires updating with more recent studies.

Author response: We updated the manuscript with more recent studies. We updated the literature with several recent studies (2024–2025) on bachelor/master theses in nursing/health sciences, including a prospective cohort on evidence-based practice (EBP) gains during thesis writing, qualitative studies on supervision and learning, a pre–post survey on group supervision, and a 2025 qualitative study synthesizing student, tutor and clinician perspectives. We found a study from Brazil about the dissemination of physiotherapy theses, etc. These additions strengthen the rationale and contextualize our findings within the latest evidence.

3. The study’s aim is well-articulated, targeting a significant gap - Publication outcomes and research trends in Bachelor and Master theses in health sciences in Croatia.

Author response: We are grateful to the reviewer for highlighting that the study’s aim is clearly presented and addresses a significant research gap. This acknowledgement reinforces the value of our work.

4. Better to separate the Strengths and limitations of this study. For instance, the following is not clear if it is strength or limitation since the authors searched only 3 databases including Google and Google Scholar and did not include CINAHL and the search strategy was not clear and systematic.

Author response: The section “Strengths and limitations of this study” with bullet points is written in line with the journal instructions. However, to address this point, we have now revised the Discussion to include a separate section on strengths and limitations there. We believe that our use of the 3 information sources could be seen as both a strength and a limitation. As a strength, on one side, we used 3 different information sources to try to find publications. On the other side, it could be indeed seen as a limitation that we did not use more information sources such as CINAHL so our search for publications may not be considered systematic. This was now all transparently presented in the Discussion.

5. In the introduction pages 68 and 69: you are limiting the interest of bachelor students to nursing research although in the title you mentioned “health sciences”. Quote: “writing a bachelor thesis is associated with greater interest and a favorable attitude towards nursing research and development.”

Author response: We concur that we wrote about nursing in the opening section of the Introduction. We have now cited literature about the effect of theses in study programs other than nursing, in health sciences.

6. Create a concise results table:

Tighten your results with one clear table right after the narrative results so readers can scan the data fast.

Author response: We agree that a single, concise summary table will help readers scan the key outcomes rapidly. We have added one at-a-glance results table that should be placed in the manuscript immediately after the narrative Results. This is now Table 6. The existing detailed tables remain available for depth.

7. Open the discussion with the main finding, then spell out its impact

7.1. One sentence: restate the top result.

Author response: We have now changed the beginning of the Discussion. The opening sentence is now revised into: “In a national cohort of 9,861 health-sciences theses, most were literature reviews rather than original studies, especially at the bachelor level, and only 2.8% of theses yielded a peer-reviewed journal article.”

7.2. Two to three sentences: explain how it shifts current evidence, guides future studies, EBP or changes bedside care.

Author response: We added the following text to accommodate this suggestion: “This quantifies a substantial evidence-to-publication gap across Croatian health sciences programs and provides a baseline for educational and policy reforms. Prior work links thesis-based original research with stronger evidence-based practice competencies; thus, curricula that increase original research theses, strengthen supervision and research methods training, and incorporate explicit dissemination plans may translate to better bedside decision-making. Future studies should investigate whether changes, such as requiring original research for all types of theses and structured supervision, affect the dissemination output of health sciences theses.”

7.3. Link results to implications for research and practice.

Author response: We added the following text, in line with this suggestion: “These results point to specific actions for programs, requiring original research conducted for bachelor and master theses, strengthening structured supervision and writing support, and planning dissemination in a scholarly journal from the outset. On the practice side, more original, patient- and service-focused student projects should translate into stronger EBP skills and locally relevant evidence that can be incorporated into protocols and quality-improvement efforts.”

8. Rewrite the conclusion so it mirrors the results and discussion

Summarize the principal outcome and its practical meaning.

Echo any limits you noted earlier and close with a clear take-home message. Conclusion must reflect the evidence presented.

Author response: As suggested, we have now rewritten the conclusion. The new version of the conclusion is: “In a national cohort of 9,861 health sciences theses, 70% were based on a literature review (70%), while the rest described original studies, and results of only 2.8% theses were disseminated as a journal article. Our estimates should be interpreted in light of several limitations, as the mentor survey response was 40%, some defended theses were not available in electronic form, and our publication search was limited to three information sources. These results quantify a substantial thesis-to-publication gap. Sustained improvement will require a deliberate shift from essay-based theses to empirical studies, coupled with systematic dissemination in scholarly journals, so that student research can credibly inform clinical practice and health policy.”

8. This paper has potential but needs a stronger analytical voice and updated evidence base to make a meaningful contribution.

Author response: We appreciate that the reviewer recognized potential in our manuscript. We hope that the revisions made will be considered adequate.

Reviewer #2:

1. There are several grammatical errors in the manuscript. The authors should use a tool such as Grammarly to help them make corrections, such as Bachelor's and Master's theses in the manuscript.

Author response: We have checked the entire text of the manuscript for English language editing, and made multiple language revisions.

2. The Discussion was particularly shallow for the huge data they gathered in 5 tables. There should have been at least one major takeaway from each of the tables. This is my major criticism of the manuscript.

Author response: We appreciate this major point. We revised the Discussion to include five concise, table-anchored takeaways, one for each of Tables 1–5. Now, the narrative explicitly interprets the core signals in the data. Each takeaway is linked to an implication for education, research training, or dissemination.

Specifically, the following was added to the Discussion, to introduce specific takeway for each of the Tables 1-5:

For Table 1: “Of note, our sample of theses was dominated by professional/polytechnic programs (71%) and bachelor-level theses (81%), with the largest single institutional contributor accounting for nearly half of all theses; programs in nursing (57%) and physiotherapy (21%) predominate. This composition helps explain the overall skew toward non-experimental theses work and should be considered when generalizing beyond Croatian health sciences programs.”

For Table 2: “Across all analyzed theses in our sample, literature reviews far outnumbered original studies (70% vs 30%). Where designs were stated, reporting was often incomplete (“not specified” 70%); where we could classify, cross-sectional designs dominated (73%) and truly experimental work was rare. This pattern signals both a capacity gap for primary research and a reporting gap that programs could address in future interventions.”

For Table 3: “An important interpretive point is the alignment between the level of education and the thesis type. At the master's level, theses were mostly based on original research (66%), whereas bachelor's theses were mostly literature reviews (79%). Yet even within original-research theses, cross-sectional designs predominated for both levels, indicating an opportunity to foster longitudinal, interventional, or mixed-methods projects, particularly at the master’s level, where student capacity is higher.”

For Table 4: “Among mentors responding (40%), only 4.5% reported a published article; submissions in progress were rare (0.9%). The leading barriers were student lack of interest (54%) and insufficient thesis quality (19%), pointing to modifiable factors (motivation, writing/statistical support, authorship planning) rather than non-modifiable constraints.”

For Table 5: “Overall, among 2.8% theses that led to a journal article, students were first authors in 66% and mentors last authors in 42%. Also, it should be emphasized that 83% of outputs in articles appeared in Croatian journals, with potentially limited international reach. Strengthening English-language editing, journal targeting, and indexing coverage could raise discoverability and impact.”

3. The suggestion by the authors that the theses can be replaced by an examination is counterproductive. It is bad enough that many of their students are merely carrying out literature reviews instead of doing laboratory or field work; the authors should realize that research training is also an important component of university education.

Author response: We concur with the Reviewer, and we revised the discussion in that respect. We have now simply mentioned that one of the institutions in our study replaced the bachelor's thesis with an exam, but that this may not be the ideal solution because research training is an essential component of university education.

We hope that the revised manuscript will be adequate.

Sincerely,

Livia Puljak and co-authors

---

## [Editor Report · Decision Letter 1]

9 Oct 2025

Publication outcomes and research trends in bachelor’s and master’s theses in health sciences in Croatia: retrospective cohort study and survey of mentors

PONE-D-25-36339R1

Dear Dr. Livia Puljak,

We’re pleased to inform you that your manuscript has been judged scientifically suitable for publication and will be formally accepted for publication once it meets all outstanding technical requirements.

Kind regards,

Iskra Alexandra Nola

Academic Editor

PLOS ONE
---

## [Editor Report · Acceptance letter]

PONE-D-25-36339R1

PLOS ONE

Dear Dr. Puljak,

I'm pleased to inform you that your manuscript has been deemed suitable for publication in PLOS ONE. Congratulations! Your manuscript is now being handed over to our production team.

Kind regards,

on behalf of

Dr. Iskra Alexandra Nola

Academic Editor

PLOS ONE